# Unveiling the High Diversity of Clones and Antimicrobial Resistance Genes in *Escherichia coli* Originating from ST10 across Different Ecological Niches

**DOI:** 10.3390/antibiotics13080737

**Published:** 2024-08-06

**Authors:** Maxsueli Aparecida Moura Machado, Pedro Panzenhagen, Cesar Lázaro, Miguel Rojas, Eduardo Eustáquio de Souza Figueiredo, Carlos Adam Conte-Junior

**Affiliations:** 1Food Science Program (PPGCAL), Chemistry Institute (IQ), Federal University of Rio de Janeiro (UFRJ), Rio de Janeiro 21941-909, Brazil; conte@iq.ufrj.br; 2Center for Food Analysis (NAL), Technological Development Support Laboratory (LADETEC), Federal University of Rio de Janeiro (UFRJ), Rio de Janeiro 21941-598, Brazil; 3Laboratory of Advanced Analysis in Biochemistry and Molecular Biology (LAABBM), Department of Biochemistry, Federal University of Rio de Janeiro (UFRJ), Rio de Janeiro 21941-909, Brazil; 4Oswaldo Cruz Institute, Rio de Janeiro 21040-900, Brazil; 5Laboratory of Veterinary Pharmacology and Toxicology, Faculty of Veterinary Medicine, National University of San Marcos, Lima 03-5137, Peru; clazarod@unmsm.edu.pe; 6Laboratory of Immunology, Faculty of Veterinary Medicine, National University of San Marcos, Lima 03-5137, Peru; mrojasm2@unmsm.edu.pe; 7Animal Science Program (PPGCA), Federal University of Mato Grosso (UFMT), Cuiabá 78060-900, Brazil; eduardo.figueiredo@ufmt.br; 8Nutrition, Food and Metabolism Program (PPGNAM), Federal University of Mato Grosso (UFMT), Cuiabá 78060-900, Brazil

**Keywords:** *E. coli*, sequence types, multidrug resistance, high pathogenicity island, whole genome sequencing

## Abstract

In this pioneering in silico study in Peru, we aimed to analyze *Escherichia coli* (*E. coli*) genomes for antimicrobial resistance genes (ARGs) diversity and virulence and for its mobilome. For this purpose, 469 assemblies from human, domestic, and wild animal hosts were investigated. Of these genomes, three were *E. coli* strains (pv05, pv06, and sf25) isolated from chickens in our previous study, characterized for antimicrobial susceptibility profile, and sequenced in this study. Three other genomes were included in our repertoire for having rare cgMLSTs. The phenotypic analysis for antimicrobial resistance revealed that pv05, pv06, and sf25 strains presented multidrug resistance to antibiotics belonging to at least three classes. Our in silico analysis indicated that many Peruvian genomes included resistance genes, mainly to the aminoglycoside class, ESBL-producing *E. coli*, sulfonamides, and tetracyclines. In addition, through Multi-locus Sequence Typing, we found more than 180 different STs, with ST10 being the most prevalent among the genomes. Pan-genome mapping revealed that, with new lineages, the repertoire of accessory genes in *E. coli* increased, especially genes related to resistance and persistence, which may be carried by plasmids. The results also demonstrated several genes related to adhesion, virulence, and pathogenesis, especially genes belonging to the high pathogenicity island (HPI) from *Yersinia pestis*, with a prevalence of 42.2% among the genomes. The complexity of the genetic profiles of resistance and virulence in our study highlights the adaptability of the pathogen to different environments and hosts. Therefore, our in silico analysis through genome sequencing enables tracking the epidemiology of *E. coli* from Peru and the future development of strategies to mitigate its survival.

## 1. Introduction

Antimicrobial resistance (AMR) is one of the main public health problems of the current era, affecting humans, animals, the environment, and, consequently, the world economy [1]. The practice of using antibiotics in human treatment has also changed in the post-pandemic period of COVID-19, increasing the World Health Organization’s alertness regarding the rational use of antibiotics for patients with the disease [1]. In this regard, four infectious syndromes are often associated with bacteria (bloodstream, gastrointestinal, gonorrhea, and urinary tract). Bloodstream and urinary tract infections were the most reported until 2020, with *E. coli* being the most frequent pathogen responsible for both, resulting in many deaths around the world [1,2].

The most common antibiotics used to treat infections around the world include first- and second-line antibiotics such as ampicillin, co-trimoxazole, and fluoroquinolones. They are also the most associated with *E. coli* resistance rates [1]. Likewise, in modern animal production, the use of antimicrobials is frequent and is likely to increase by up to 67% by 2030 [3]. Even drugs of last choice for treating infections caused by *E. coli*, such as colistin, can be overcome by the bacteria when the transfer of plasmid-mediated mobile-colistin resistance genes (*mcr*) occurs. In this regard, chickens and pigs are considered the main reservoirs of *mcr*-mediated colistin-resistant *E. coli* globally [4].

The sheer quantity of resistance genes found in the *E. coli* genome outweighs that of any other bacteria [5]. According to Raymond et al. [5], the robustness of the *E. coli* accessory genome in harboring antimicrobial resistance genes makes the bacteria’s machinery a frequent target for the transfer of mobile elements that subsequently integrate into the chromosome. Pandemic strains of *E. coli* from a wide variety of sources, from food to human environments, have a significant role in the epidemiology of the bacteria [6,7,8,9]. Most of the widespread *E. coli* strains with resistance genes, identified as linages ST10, ST69, ST73, ST95, ST131, ST38, ST58, ST155, ST156, ST167, ST393, ST405, ST410, ST648, ST617, ST665, ST744, and ST998, have been observed in both human and animal hosts [6,9,10,11,12,13]. The ability of these clonal strains to share resistance and virulence genes also influences commensal populations that become pathogenic due to the acquisition of mobile genetic material [14]. In this sense, pathogenic *E. coli* serotypes such as O157:H7 harboring virulence genes are the major cause of human infections due to consuming contaminated food [15]. Furthermore, by acquiring mobile genetic material such as phages and plasmids, *E. coli* can also acquire pathogenicity islands like those in *Yersinia strains*. The high-pathogenicity island (HPI) from *Yersinia* encompasses multiple genetic markers responsible for encoding yersiniabactin-mediated iron uptake, which increases the metabolism of the host such as chickens and rodents. Consequently, it causes septicemia in mammals and birds [16,17,18,19].

In order to investigate the antimicrobial resistance and virulence profiles of *E. coli* from Peru, we performed a microbiological analysis and whole genome sequencing (WGS) in strains isolated from chickens. In addition, we applied the WGS analysis as a comprehensive study in order to determine the diversity of virulence and resistance in the entire *E. coli* genomes from Peru.

## 2. Results

### 2.1. Antimicrobial Resistance among E. coli Genomes from Peru

The phenotypic analysis of antimicrobial resistance in our three strains from chicken legs (pv05, pv06, and sf25) showed that they were all multi-drug resistant, with resistance to more than three classes of antibiotics. The strains exhibited resistance to the macrolide (azithromycin, erythromycin), tetracycline (tetracycline), fluoroquinolone (ciproxoxacillin), and penicillin (amoxicillin) classes. Exceptionally, the sf25 strain demonstrated resistance to other classes of antibiotics, such as aminoglycosides (neomycin), phenicol (fluorophenicol), and other penicillin-like antibiotics (amoxicillin-clavulonic acid; intermediate resistance).

Regarding our in silico analyses, the results revealed that over twenty genes are predicted to confer resistance to numerous classes of broad-spectrum antibiotics used in human and animal practice (Appendix A). Among them, we highlight the *aac*, *aad*, *ant*, and *aph* genes for resistance to aminoglycosides with a frequency reported in up to 232/469 genomes. Of the genes that confer resistance to extended-spectrum beta-lactamases (ESBL) (*blaSHV*, *blaCTX-M-55*, *blaCTX-M-65*, *blaCTX-M-90*, *blaTEM*), the *blaTEM* gene was the most frequent among the genomes (251/469). Other highly frequent genes between the genomes were *dfr* (243/469) related to dihydrofolate-trimethoprim, *flor* (194/469) to broad-spectrum antibiotics, *fos* (192/469) to fosfomycin resistance, *qnr* (121/469) and *gyrA* (142/469) to fluoroquinolones, *sul* (285/469) to sulfonamides, and *tet* (261/469) to tetracyclines (Figure 1). These genomes were predominantly isolated from clinical sources in human hosts (344/466), representing 78% of genomes, followed by domestic and wild animals.

### 2.2. The Evolution of the E. coli Accessory Genome Originating from ST10

Based on our analysis of the pan-genome generated by prokka, we identified 24,507 genes (Figure 2). Of these, 2869 belonged to core genes, and 18,856 were accessory genes. Our phylogenetic tree was divided into four clades (A, B, C, and D). We also noticed that as new lineages emerged (clades C and D), the number of accessory genes increased. Furthermore, we identified the absence of some core genes in clade D (identified by the white color in the blue part of the figure). In addition, according to Achtman’s MLST scheme, more than 180 different sequence types (STs) were identified (Appendix A). For each clade, we identified the three most frequent STs among the genomes (Figure 3). The other STs not identified in the figure were less frequent among the genomes, present in only seven or fewer.

The ST10 pandemic lineage, associated with human, animal, and environmental sources, was present in most of the genomes in clade A and in some strains in clade B. However, it was completely extinct in the genomes of clades C and D (Figure 3). Our analysis also identified three rare core genomes between Peruvian *E. coli* (cgST31870, cgST61634, cgST115873), seen only once in three other genomes on the EnteroBase platform (SRR10233444, ERR2204726, ERR1622878). Exceptionally, these cgSTs were found in our three strains of *E. coli* isolated from chickens. The ST162, originating from cgST115873, was also associated with two other *E. coli* genomes from Peru (GCA_021699765.1 and GCA_025564015.1) isolated from humans and animals.

### 2.3. Serotypes and Virulence Genes

Our analysis revealed a diversity of more than 200 serotypes, the most prevalent of which were O8, O89, O99, O9, and O153 (Appendix A). Seven serotypes mostly associated with *E. coli* infection were found among the genomes within the frequencies (except serotype O157 and O145, which was not present) O26 (11/469), O45 (4/469), O111 (4/469), O103 (1/469), O121 (1/469). Despite these serotypes being associated with foodborne *E. coli*, we did not find any of them in our chicken strains. Instead, all of them were associated with clinical sources (human and animal). Furthermore, our analysis revealed virulence genes, especially those in the high-pathogenicity island (HPI) of *Y. pestis* and *E. coli* variants (Appendix A). The HPI prevalence was 42.2% (198 out of 469 genomes), predominantly associated with the HPI *Y. pestis* variant. Among genomes with HPI genes, 172/198 showed complete HPI. Most of the genomes presented partial HPI genes. Between our chicken genomes, just the pv05 showed 92% coverage for the HPI *Y. pestis* variant.

Additional adherence factors, protectins, and the *ibeA* gene for adherent-invasive *E. coli* (AIEC) were identified among the genomes set (Appendix A). For genomes pv05, pv06, and sf25, the analyses revealed the presence of adherence-associated genes (*sslE*, *iha*, *lpfA*, *tsh*); another gene for extraintestinal pathogenic *E. coli* (ExPEC) association (*cvaC*); and a heat-stable enterotoxin, the *astA* gene, for enterohaemorrhagic *E. coli* (EHEC) and enteroaggregative *E. coli* (EAEC) [20,21,22,23,24,25].

### 2.4. Characterization of Persistence Genes and Plasmids

Fifty-one genes available on the BioSim platform (https://biosim.pt/, accessed on 31 December 2023) related to biofilm formation were screened, with their functions related to the expression of proteins for dispersion, extracellular polymeric substance (EPS), motility, and quorum sensing [26]. The number of these genes among the genomes was between nine and seventeen, independent of the source. Additional genes related to stress were identified in association with resistance to biocides (*emrE*), metals (*mer*, *pco*, *ars*, *sil*), tellurium (*ter*), and quaternary ammonium (*qacL* and *qacE*). Of these, the *emrE* (309/469), and *qacL*/*qacE* (225/469) genes were the most frequent in the entire population of this study. Regarding the mobilome/plasmids, many incompatibility groups were found, especially col-like (col156, colF, colpVC, colE10, colkp3, col3m, col8282) with a frequency of 159/469 and inc-like (incB-C-like, incF-like, incFIA, incFIB, incFII, incH-like, incI-like, incN-like, incQ-like, incX-like, incY-like) with a frequency of 367/469 (Appendix A).

## 3. Discussion

The historical issue of indiscriminate antimicrobial use in human and animal practice is currently evident and will persist in the future [3,27]. High rates of antimicrobial resistance between *Enterobacteriaceae* can be found in South American countries [27]. Poor investment in basic sanitation, combined with overcrowded populations and low levels of education among farmers, contribute to the emergence and spread of super-resistant strains [28]. These issues among producers are a situation mainly observed in Peru, where the most significant production of animal-based food is concentrated in cattle, pigs, and chickens, and most of this production is carried out by small farmers who lack knowledge about the correct use of antibiotics [28,29].

Antibiotics belonging to tetracyclines, penicillins, and aminoglycosides are frequently used by farmers in Peru, with percentages of up to 31% [28]. Our results also found a high frequency of genes for all these classes, especially tetracyclines, sulfonamides, and aminoglycosides (Figure 1). Phenotype resistance to macrolides, tetracyclines, penicillins, and aminoglycosides was identified for strains pv05, pv06, and sf25. The phenotype of resistance to more than three classes, although only identified in three strains, together with numerous genes belonging to different classes of antibiotics, highlights the problem of antimicrobial resistance in *E. coli* circulating in different hosts in Peru. Although most of the Peruvian genomes originated from human hosts (78%), the same profile for antimicrobial resistance genes in *E. coli* was seen in all the others hosts (chickens, pigs, bats, and cows using clinical sources) (Figure 1).

The high risk of these strains overcame antimicrobial barriers since several virulence and stress resistance genes, such as biocides, metals, and biofilm, were found, especially among the strains isolated from food (pv05, pv06, and sf25) (Figure 2). The high pathogenicity island (HPI) was found to have a prevalence of 42.2% amongst the genomes (Appendix A). HPI in *E. coli* has been evidenced in highly virulent *E. coli*, capable of causing bacteremia and urosepsis in animals and humans [16,17,18,19,30]. Additionally, the HPI can be found in numerous serovars and pathovars of *E. coli* [31], and the presence in humans, animals, and food, as demonstrated in our study (Appendix A), highlights the wide spread of *E. coli* from Peru.

Cross-resistance between human and animal bacteria can be also caused by the transmission of plasmids carrying antimicrobial resistance and virulence genes. The most common plasmid replicon types can harbor numerous resistance genes capable of coding for different classes of antibiotics, facilitating selection even without the selective pressure of the environment [32]. Inc-like incompatibility groups are one of the most important plasmid replicons involved in carrying antimicrobial resistance genes between bacteria, in particular, ESBL genes [32,33]. Based on our results, we found that Inc-like was the most frequent within the Peruvian genomes (367/469), coinciding with the high frequency of resistance genes from the ESBL group (251/469).

Previously, in a study by Benavides et al. [34] with *E. coli* from Peru, the ESBL resistance genes (*blaCTX-M-15*) were found in *E. coli* from livestock and bats that feed on livestock. Our global in silico analysis also found the spread of resistance and virulence genes not only between domestic and wild animals but also between *E. coli* from food and humans. Among the only three genomes of *E. coli* from food (strains pv05, pv06, and sf25), we found rare cgMLSTs (cgMLST 31870, 61634, and 115873) observed only in *E. coli* from poultry farms in Vietnam (ERR1622878), in *E. coli* isolated from humans in Brazil (ERR2204726), and in *E. coli* from an Andean bird in Chile (SRR10233444). This *E. coli* profile from Peru in different hosts suggests the spread of rare clones that may have migrated through birds or the environment (contaminated water) from other countries to Peru or vice versa.

The results of our study also demonstrated the diversity of STs present in the genomes of *E. coli* from Peru (Figure 3), which may be related to the evolution of the bacteria in the face of different selective pressures from environments. The study of sequence typing plays an important role in the epidemiology and tracking of endemic populations and can be useful for verifying specific characteristics of certain regions [35]. Endemic ubiquitous lineages such as ST10 are widespread among *E. coli* from humans, animals, plants, and the environment [12,13,36,37] and were the most frequent in the Peruvian genomes (Figure 3). In addition, the proliferation of resistance genes through plasmids may be more frequent in universal STs such as ST10 [32]. The distribution of ST10 among *E. coli* harboring resistance genes has been already elucidated [9,37]. The features of the genomes from Peru may still elucidate a pellicular characteristic: as ST10 disappeared in the population of clades C and D, the number of accessory genes increased (Figure 2).

The accessory genes present in *E. coli* from Peru exceeded 24,000 genes, while the genes present in the conserved genome (core genome) stood at just over 2800 (Figure 2). Presumably, as new lineages emerged, more accessory genes were acquired, whether through plasmids (integrons or transposons) or bacteriophages [38]. This characteristic of a large genome in *E. coli* has already been discussed and can be explained by the selective pressure of the environment suffered by the bacteria [5,35]. However, according to Robins-Browne et al. [35], the changes in the core genome and accessory genome of *E. coli* can also be explained by the emergence of new assemblies, i.e., as more genomes are sequenced, the greater the changes in genome structure. Furthermore, the low frequency of Peruvian genomes available in the NCBI platform from other sources (animals and food) reflects the relatively low investment and investigation to better elucidate the genetic characteristics of the pathogen in the country. Therefore, this study underlines the importance of in silico analysis through WGS, especially in ubiquitous populations. Determining genetic characteristics in terms of the diversity of resistance, virulence, and persistence genes is key to providing information on monitoring, controlling, and creating strategies to mitigate the evolution of *E. coli* in Peru.

## 4. Materials and Methods

### 4.1. Sample Obtention and Antibiotic Susceptibility

Three strains identified as *E. coli*, named pv05, pv06, and sf25, preserved in brain heart infusion (BHI) broth (Oxoid, Hampshire, UK) containing 15% glycerol and stored at −80 °C at the Laboratory of Veterinary Pharmacology and Toxicology of the Universidad Nacional Mayor de San Marcos, were used on this study. These strains were isolated from cuts of chicken legs in traditional markets in Lima, Peru, from one of our previous studies [39]. The strains were reactivated on MacConkey agar (MerckTM, Darmstadt, Germany) and incubated at 37 °C/24 h. 

The antimicrobial susceptibility assays used the diffusion disk technique or Kirby–Bauer test. The following antibiotics commonly used in human and animal medicine and available in our laboratory were analyzed in this study: azithromycin (15 μg), erythromycin (15 μg), tetracycline (30 μg), ciprofloxacin (5 μg), amoxicillin (10 μg), neomycin (30 μg), amoxicillin + clavulanic acid (20/10 μg), and florfenicol (30 μg). For this purpose, a colony of *E. coli* was inoculated into Mueller–Hinton broth (Condalab^®^, Madrid, Spain) and incubated at 37 °C/24 h. The bacterial growth earned a score of 0.5 on the MacFarland scale compared to a standard barium sulfate solution (Liofilchem, Roseto degli Abruzzi TE, Italy). Next, the culture was prepared and seeded on Muller–Hinton agar (Condalab^®^, Madrid, Spain) plates, and the antibiotic disks were included and incubated at 37 °C/24 h. The results were analyzed according to the Clinical and Laboratory Standards Institute (CLSI) guidelines, with sensitive strains (S) showing inhibition halos ≥ 26 mm, intermediate (I) showing halos between 23–25 mm, and resistant (R) showing a halo ≤ 22 mm [40].

### 4.2. Whole Genome Sequencing

The three *E. coli* strains previously characterized for antimicrobial resistance were sequenced in this study. A single colony was used for total DNA extraction using the Wizard Genomic DNA Purification kit (Promega^®^, Madison, WI, USA). The NEBNext1 Library Preparation Kit from New England Biolabs Inc., Ipswich, MA, USA, was utilized to prepare sequencing libraries, following the manufacturer’s protocol. Subsequently, sequencing was performed on the Illumina HiSeq 2500 platform (Illumina Inc., San Diego, CA, USA) to achieve a minimum coverage of 100×. The quality assessment was carried out using FastQC v0.11.5, and the data were processed through fastp v0.23.4 [41]. For de novo assembly, Unicycler’s standard pipeline (https://github.com/rrwick/Unicycler, accessed on 31 December 2023) was employed to produce superior quality assemblies compared to traditional tools [42]. Assembly metrics were evaluated using QUAST v5 [41]. These sequences were deposited in NCBI GenBank at BioProject PRJNA1100992.

Lastly, to perform an in-depth analysis of *E. coli* from Peru, we downloaded all the genomes (*n* = 463) available on the Pathogen Detection platform of the National Center for Biotechnology Information (NCBI (https://www.ncbi.nlm.nih.gov/pathogens/, accessed on 23 October 2023)). These genomes were isolated from different sources and hosts between 1983 and 2021.

### 4.3. In Silico Genotype Profiling

For antimicrobial resistance and plasmids profiles, we conducted scans against the ResFinder [43], PointFinder [44], and PlasmidFinder [45] databases to identify antimicrobial resistance genes and plasmids using the STARAMR 0.7.1 software from https://github.com/phac-nml/staramr (accessed on 31 December 2023). The software was employed to analyze bacterial genome contigs, producing a comprehensive report that summarized the detected antimicrobial resistance genes and plasmids. Additionally, it predicted drug resistance based on the identified resistance genes. Our scanning process adhered to stringent criteria, with a minimum DNA identity threshold of 95% and a minimum DNA coverage of 60% applied to all genome alignments against the databases. We used the ‘mlst’ software developed by Torsten Seemann (Seemann T, mlst Github: https://github.com/tseemann/mlst, accessed on 31 December 2023). The analysis was performed using the conventional MLST server, PubMLST (https://cge.food.dtu.dk/services/MLST/, accessed on 31 December 2023), which relies on whole genome sequence (WGS) data to determine bacterial sequence types (STs) [46]. For the identification of typing schemes and STs, we specifically chose seven housekeeping genes: *aroC* (chorismate synthase), *dnaN* (DNA polymerase III beta subunit), *hemD* (uroporphyrinogen III cosynthase), *hisD* (histidinol dehydrogenase), *purE* (phosphoribosylaminoimidazole carboxylase), *sucA* (alpha-ketoglutarate dehydrogenase), and *thrA* (aspartokinase + homoserine dehydrogenase).

In order to identify the Core Genome Multi-Locus Sequence Type (cgMLST), we employed the open-access cgMLSTFinder software provided by the Center for Genomic Epidemiology (https://bitbucket.org/genomicepidemiology/cgmlstfinder/src/master/, accessed on 31 December 2023). This software utilizes rapid and precise K-mer alignment of reads [47] against EnteroBase to establish cgMLST schemes based on alleles present in the 3002 Salmonella genes [48].

The Biofilms Structural Database—biosim (https://biofilms.biosim.pt, accessed on 31 December 2023), with 51 genes, and AMRFinderPlus (NCBI) were employed using Seemann T.’s ABRicate software (v1.0.0, https://github.com/tseemann/abricate, accessed on 31 December 2023) for biofilm and virulence genes, respectively. Our screening criteria involved a minimum per base identity of 95% and 60% coverage. The intentional setting of the coverage threshold aimed at ensuring comprehensive screening, accounting for genes potentially located on the edge of a contig or spanning multiple contigs due to potential non-perfect assembly.

The prevalence of a high-pathogenicity island (HPI) was analyzed through two variants based on previous studies [16,17] conducted on *Yersinia pestis* (*Y. pestis*) (accession number AL590842) and *E. coli* (accession number AY233333). Partial genomes were defined by a cutoff of 11–65%, while complete genomes were defined by a cutoff of 80–100%.

### 4.4. Pan-Genome Analysis

For the pan-genome analysis, we used 469 *E. coli* genomes, 466 of which were *E. coli* genomes from Peru plus three *E. coli* assemblies selected by PubMLST because they contained the same cgMLST as the *E. coli* strains isolated from chickens (pv05, pv06, and sf25). For this analysis, we used Panaroo v1.3.4 [49]. The whole genome alignment was conducted with a minimum identity percentage of 95% for blastp, requiring a gene’s presence in at least 99% of the isolates to be deemed central. Standardized genome annotation was ensured using prokka v1.14.5 [50]. Visualization of the pan-genome was accomplished through post-processing scripts provided by Roary. To construct a maximum likelihood phylogeny from the central genome alignment, IQtree v2.0.3 [51] was employed. This involved using a GTR+F+I+G4 substitution model with 1000 bootstrap replications for node support. Complementary annotation was carried out using emapper v2.0 [52] with default parameters, relying on orthology data from eggnog [53]. Diamond protein alignment [53] facilitated sequence searches. For interactive visualization of phylogenetics, metadata, and gene content, the Phandango browser application [54] was utilized. The genomes were classified into four clades (A, B, C, and D) based on the presence of unique genes shared by all genomes within each cluster. This means that genomes within the same clade possessed distinct genes not found in other clades, suggesting a closer evolutionary relationship. The distribution of genomes among the clades was as follows: Clade A (*n* = 127), Clade B (*n* = 84), Clade C (*n* = 132), and Clade D (*n* = 126).

## 5. Conclusions

The present study revealed many antimicrobial resistance genes related to aminoglycosides, ESBL-producing *E. coli*, sulfonamides, and tetracycline classes among Peruvian *E. coli* genomes from human, domestic, and wild hosts. In addition, three genomes sequenced in this study (pv05, pv06, and sf25) isolated from chickens presented a multi-drug resistance phenotype to more than three classes of antibiotics. Most of Peru’s genomes harbored the endemic ST10 clone (77 out of 469 genomes). As new clones have emerged in *E. coli* from Peru, the ability to acquire accessory genes has increased. Also, rare clones (cgMLST 31870, 61634, and 115873) were seen for the first time in the country on those chicken *E. coli* genomes. Moreover, besides antimicrobial resistance, the pathogenicity of these strains was associated with the presence of genes identified in the high pathogenicity island (HPI), which was found in 42.2% of the genomes. Our comprehensive analyses also showed that biocide, metal, and biofilm genes were more present in foodborne *E. coli* genomes (pv05, pv06, and sf25), suggesting the pathogen’s ability to adapt to conditions that can be found in food processing and distribution facilities. Nevertheless, the low availability of *E. coli* genomes in Peru may hinder a more accurate in silico assessment regarding the prevalence of this pathogen and the primary source(s) of its isolation. Therefore, our study emphasizes the importance of WGS in identifying resistance, virulence, and persistence genes in *E. coli*, which is essential for monitoring, managing, and mitigating the evolutionary processes of the bacteria.

## Figures and Tables

**Figure 1 antibiotics-13-00737-f001:**
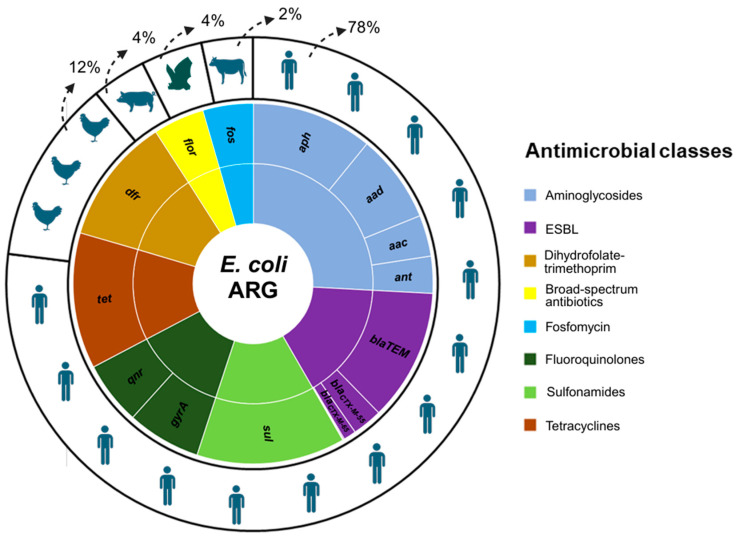
Distribution of the main antimicrobial resistance genes and classes identified in *E. coli* from Peru originating from human and animal hosts. Caption: the small circle shows the frequency of antimicrobial genes and their antibiotics classes in *E. coli* genomes (*n* = 469), and the larger circle shows the percentage of the genomes in the hosts.

**Figure 2 antibiotics-13-00737-f002:**
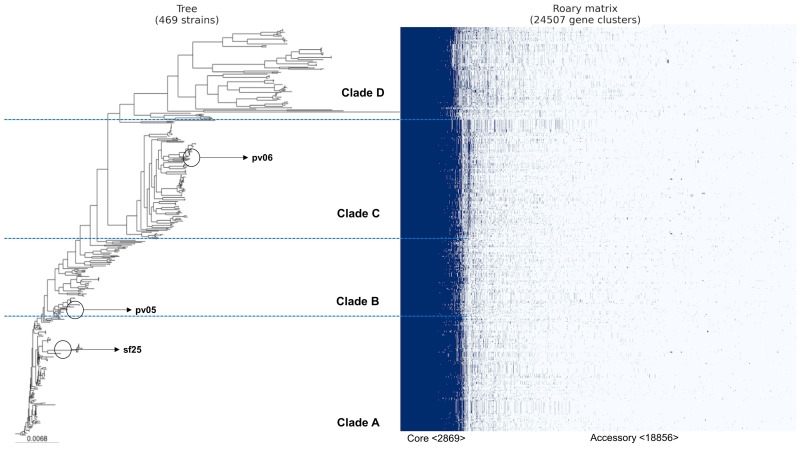
Peruvian *E. coli* pan-genome based on the presence/absence of core and accessory genes using ROARY. The classification of genomes into clades was carried out by zooming in on subsets of genomes equidistant from the ancestral genome.

**Figure 3 antibiotics-13-00737-f003:**
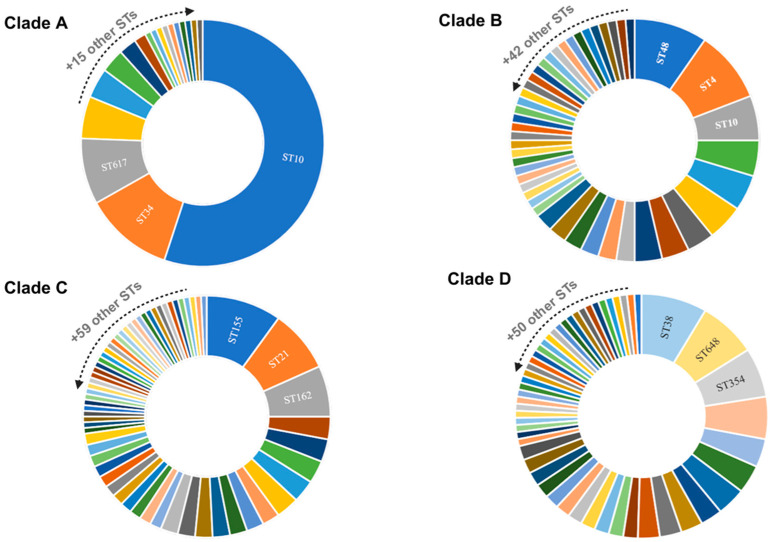
Principal sequence types (STs) identified in *E. coli* genomes. Caption: Distribution of the most frequent sequence types (STs) among genomes. The clades indicate the subsets of genomes visualized through the interactive phylogenetic tree. Clade A (*n* = 127), clade B (*n* = 84), clade C (*n* = 132), and clade D (*n* = 126).

## Data Availability

The datasets generated during the current study are available in the NCBI repository under the BioProject access number PRJNA1100992.

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
