# Peer review of "Unveiling the High Diversity of Clones and Antimicrobial Resistance Genes in Escherichia coli Originating from ST10 across Different Ecological Niches"

_antibiotics, 2024, doi:10.3390/antibiotics13080737_

Round 1

Reviewer 1 Report

Comments and Suggestions for Authors

The authors of the study conducted wgs on 3 strains of E. coli that were isolated for an earlier study from cuts of chicken legs in a market in Peru. The aim of this study was to identify antimicrobial resistance genes in the 3 strains. The authors find that the 3 strains are phenotypically resistant to more than 3 classes of antibiotics. 

The authors then extended the study to include 469 genomes of bacteria (predominantly E. coli) isolates from 1983 to 2021 from Peru. 

While the idea behind this study is interesting, the paper seems to lack focus and doesn't draw any interesting conclusions. For instance, there is no description of what the origin of these different strains are apart from them all being from Peru. Furthermore, while the study claims to have begun to analyze the 3 strains from chickens, those are the only 3 strains with a known source, and the rest could all be from human patients. 

Comments on the Quality of English Language

There are a lot of plural words incorrectly used throughout the text. 

Author Response

The authors of the study conducted wgs on 3 strains of E. coli that were isolated for an earlier study from cuts of chicken legs in a market in Peru. The aim of this study was to identify antimicrobial resistance genes in the 3 strains. The authors find that the 3 strains are phenotypically resistant to more than 3 classes of antibiotics. The authors then extended the study to include 469 genomes of bacteria (predominantly E. coli) isolates from 1983 to 2021 from Peru. While the idea behind this study is interesting, the paper seems to lack focus and doesn't draw any interesting conclusions. For instance, there is no description of what the origin of these different strains are apart from them all being from Peru. Furthermore, while the study claims to have begun to analyze the 3 strains from chickens, those are the only 3 strains with a known source, and the rest could all be from human patients.

A: Dear reviewer, thank you for your comments. We would like to clarify point by point. I) regarding the objectives and conclusion, we have changed these points for better clarification of the study in lines 38-41 and 379-387. II) “For instance, there is no description of what the origin of these different strains are apart from them all being from Peru”: All the genomes analyzed were from Peru since the aim of this manuscript was to carry out an in silico analysis of E. coli from the country. We already modified the goal to clarify this point (lines 84-88). The assemblies were collected by the pathogen detection platform, and the host and isolation source information has been added in supplementary table 1. In addition, in Figure 1, we have listed all the hosts from which these genomes were isolated. III) In relation to genomes hosts and isolation source, most of the genomes have humans as their host, and many can be traced back to the source of isolation in table S1. However, since this information is filled in by researchers who have deposited the sequences, many genomes have no known source of isolation, which makes it difficult to fill the information correctly in our table. However, in our discussion, we mainly consider the different hosts and highlight our three sequenced isolates as the only ones described as originating from food (pv05, pv06, sf25). This information has been added and clarified in subtopic 4.2 of material and methods.

There are a lot of plural words incorrectly used throughout the text.

A: Thank you. We proofread the entire manuscript for language appropriateness.

Reviewer 2 Report

Comments and Suggestions for Authors

Unveiling the high diversity of clones and antimicrobial resistance genes in Escherichia coli originating from ST10 across different ecological niches

Major comments

1. The author used the Illumina sequencing platform, which provides short-read sequences, and then utilized Unicycler for genome assembly. However, Unicycler is generally used for hybrid genome assembly. For short-read assembly, tools such as SPAdes or Velvet might be more appropriate for processing Illumina short-read sequences

2. What criteria or methods did the author use to divide the tree in Figure 2 into four clades? Please provide a detailed explanation of the methodology used for clustering the tree

3. The author should be careful when using pangenome analysis to classify the STs of bacteria. Pangenome trees are rooted using maximum likelihood methods, which are highly sensitive to sequence recombination and mutations. This sensitivity can introduce biases, which leads to the clustering of closely related STs into different clades and misrepresenting their true genetic relationships

4. The author identifies many types of mobile genetic elements, however, there is no discussion about them. MGEs are important factors that facilitate the spread of antimicrobial resistance and virulence-associated genes. It would improve the manuscript to add a discussion on the significance of MGEs in your bacterial isolates

5. Please discuss more about the correlation between AST results and the antimicrobial resistance genes identified in your study. This will provide a more comprehensive understanding of the resistance mechanisms in the bacterial isolates.

Minor comments

Check the correct usage of specific terms. For example, "in silico" and "de novo" should be italicized, as well as the names of genes and bacteria. Make sure that all terminology is correctly formatted throughout the manuscript

Author Response

Comments to the Author:

  1. The author used the Illumina sequencing platform, which provides short-read sequences, and then utilized Unicycler for genome assembly. However, Unicycler is generally used for hybrid genome assembly. For short-read assembly, tools such as SPAdes or Velvet might be more appropriate for processing Illumina short-read sequences

A: Dear reviewer, thank you for your comments and suggestion. We used unicycler as it is more accurate, complete and cost-effective. For bacteria, the software automatically uses SPAdes for denovo assembly from illumina fastq files. Unicycler can produce superior assemblies, as well as features and an automated pipeline. To make this explain clear, we added some sentences on methodology about our choice on lines 300-302.

  1. What criteria or methods did the author use to divide the tree in Figure 2 into four clades? Please provide a detailed explanation of the methodology used for clustering the tree

A: For sure, you are correct. I added this part on lines 361-366.

  1. The author should be careful when using pangenome analysis to classify the STs of bacteria. Pangenome trees are rooted using maximum likelihood methods, which are highly sensitive to sequence recombination and mutations. This sensitivity can introduce biases, which leads to the clustering of closely related STs into different clades and misrepresenting their true genetic relationships

A: Yes, you are absolutely correct. To avoid such biases, we could have used a core SNP matrix to construct the ML phylogeny. On the other hand, we were careful to classify the assemblies according to the interactive visualization of the tree and by considering the compiled metadata. This approach allowed us to verify the frequency of ST10 in the genomes and understand the extent to which its genetic distance has evolved as new assemblies from Peru have emerged. In our manuscript, we cited the study by Fuga et al. (2022), who also performed an in-depth analysis of representative pandemic clones with significant public health relevance. Our analysis did not classify ST clades using the pangenome likelihood method but instead provided a straightforward classification of equidistance between genomes in relation to the ancestor. We clarified this point in lines 361-366 and in the legend of Figure 2.

  1. Fuga, F.P. Sellera, L. Cerdeira, F. Esposito, B. Cardoso, H. Fontana, Q. Moura, A. Cardenas-Arias, E. Sano, R.M. Ribas, A.C. Carvalho, M.C.B. Tognim, M.M.C. de Morais, A.J.P.G. Quaresma, Â.P. Santana, J.N. Reis, M. Pilonetto, E.C. Vespero, R.R. Bonelli, A.M.F. Cerqueira, T.C.M. Sincero, N. Lincopan, WHO Critical Priority Escherichia coli as One Health Challenge for a Post-Pandemic Scenario: Genomic Surveillance and Analysis of Current Trends in Brazil, Microbiol. Spectr. 10 (2022). https://doi.org/10.1128/spectrum.01256-21.

  1. What's the optimized amount of E. coli for the study?

A: Dear reviewer, if you are referring to the amount of E. coli standardized for the phenotypic disk diffusion assay, we follow the standards recommended by the Clinical and Laboratory Standards Institute (CLSI) guidelines, which recommend that the concentration of bacteria for the test should be 0.5 on the MacFarland scale which generally refers to 8 log CFU/ml. We mention these details in lines 281-287.

  1. The author identifies many types of mobile genetic elements, however, there is no discussion about them. MGEs are important factors that facilitate the spread of antimicrobial resistance and virulence-associated genes. It would improve the manuscript to add a discussion on the significance of MGEs in your bacterial isolates

A: Thank you. We added more discussion about this point on lines 218-226, 245-246, 252-256.

  1. 5. Please discuss more about the correlation between AST results and the antimicrobial resistance genes identified in your study. This will provide a more comprehensive understanding of the resistance mechanisms in the bacterial isolates.

A: Thank you. Some clarifications were made about it on lines 202-208.

Minor comments

Check the correct usage of specific terms. For example, "in silico" and "de novo" should be italicized, as well as the names of genes and bacteria. Make sure that all terminology is correctly formatted throughout the manuscript

A: Thank you, all these aspects were checked and corrected on the manuscript.

Reviewer 3 Report

Comments and Suggestions for Authors

The authors investigated the diversity of antimicrobial resistance (AMR) genes in E. coli ST10 strains from different ecological niches in Peru. By analyzing 469 E. coli genomes, the researchers identified a high diversity of AMR genes and phylogenetic clones, highlighting significant public health implications. The findings emphasize the widespread presence of resistance genes across various hosts and environments, underscoring the need for continuous monitoring and effective strategies to combat the spread of AMR.

Major Suggestions:

  1. Clarify Methodological Details:

a.    It would be helpful to provide specific criteria for selecting the 469 assemblies. Explain the reasons for choosing these genomes and how they represent the broader population.

b.    Elaborate on the sequencing technologies and bioinformatics tools used. Describe the quality control measures taken during sequencing and data analysis to enhance reproducibility.

c.    Detail the specific software, algorithms, and parameters used in the in-silico analysis. This will help others replicate the study and assess the robustness of the findings.

  1. Enhance Data Presentation and Interpretation: Use more detailed figures and tables to summarize key findings, such as the distribution of AMR genes across different clades and hosts. Ensure that all visual aids are clearly labeled and include legends that explain the data succinctly.
  2. Expand Discussion on Implications and Future Research: I think one of the interesting parts to emphasize is discussing in more detail how the findings about AMR genes in E. coli impact public health policies in Peru and globally. Include potential strategies for mitigating the spread of resistant strains.

Minor Suggestion:

      Please improve the resolution of the figures.

Author Response

The authors investigated the diversity of antimicrobial resistance (AMR) genes in E. coli ST10 strains from different ecological niches in Peru. By analyzing 469 E. coli genomes, the researchers identified a high diversity of AMR genes and phylogenetic clones, highlighting significant public health implications. The findings emphasize the widespread presence of resistance genes across various hosts and environments, underscoring the need for continuous monitoring and effective strategies to combat the spread of AMR.

A: Thank you for your comments and questions, we clarified point by point.

  1. Clarify Methodological Details:
  2. It would be helpful to provide specific criteria for selecting the 469 assemblies. Explain the reasons for choosing these genomes and how they represent the broader population.

A: We clarified this point on lines 305-309.

  1. Elaborate on the sequencing technologies and bioinformatics tools used. Describe the quality control measures taken during sequencing and data analysis to enhance reproducibility.

A: We had clarified how the in silico analyses were carried out and the parameters used. We have highlighted these points in the lines 312-315, 318-324, 330-332, 335-338, 344-345, 350-356.

  1. Detail the specific software, algorithms, and parameters used in the in-silico analysis. This will help others replicate the study and assess the robustness of the findings.

A: These points were already highlights on 312-315, 318-324, 330-332, 335-338, 344-345, 350-356.

  1. Enhance Data Presentation and Interpretation: Use more detailed figures and tables to summarize key findings, such as the distribution of AMR genes across different clades and hosts. Ensure that all visual aids are clearly labeled and include legends that explain the data succinctly.

A: Thank you. We checked all figures and titles and the changes were made and highlights.

  1. Expand Discussion on Implications and Future Research: I think one of the interesting parts to emphasize is discussing in more detail how the findings about AMR genes in E. coli impact public health policies in Peru and globally. Include potential strategies for mitigating the spread of resistant strains.

A: Thank you. We have added these points with special emphasis on the lines: 37-40, 202-208, 218-226, 252-255, 259-266.

Minor Suggestion:

Please improve the resolution of the figures.

A: Dear, we have checked the resolution of the figures, and they all have 300 dpi according to journal guidelines. We have also attached them if you want to see them outside the manuscript.

Round 2

Reviewer 1 Report

Comments and Suggestions for Authors

While the authors have made efforts to address some of my concerns, the overall manuscripts still lacks focus. 

Reviewer 2 Report

Comments and Suggestions for Authors

The authors have addressed all my concerns; I have no further questions.